# Assembly Solution for Modular Buildings: Development of an Automated Connecting Device for Light-Framed Structures

**Laurence Picard [1], Pierre Blanchet [2]** and **André Bégin-Drolet [1,*]**

1   Department of Mechanical Engineering, Laval University, Quebec, QC G1V 0A6, Canada;
    laurence.picard.3@ulaval.ca
2   Department of Wood and Forest Sciences, Laval University, Quebec, QC G1V 0A6, Canada;
    pierre.blanchet@sbf.ulaval.ca
*   Correspondence: andre.begin-drolet@gmc.ulaval.ca

**Abstract:** The prefabricated construction industry, also known as off-site construction, has been operating in North America for several years now and differs from traditional construction in its much shorter project timelines, lower costs, and increased build quality. However, the lack of a suitable and efficient assembly solution has been identified by many as a barrier to the use of off-site construction for larger buildings. To maximise the benefits of off-site manufacturing for multistorey buildings, an automated connection solution is presented in this paper. A new plug-in self-locking device was developed according to the following product design phases: on-site observations, definition of the problem and product specifications, solution generation, prototyping, fabrication, and testing. The plug-in self-locking device allows the assembly process to be accelerated by eliminating the fastening steps and a higher completion of modules off-site to be achieved. The design bears the compressive, tensile, and shear loads and contributes to the load path of the building.

**Keywords:** assembly solution; automated connection; construction productivity; modular connection; modular buildings; plug-in device; self-locking mechanism

## 1. Introduction

### 1.1. Background

A recent study published in May 2021, led by Braham and Homsy in the province of Quebec, Canada, identified the construction field as undergoing the second most severe labour shortage among all industries [1]. Prior to the COVID-19 pandemic, the labour deficit had already been identified as a major problem directly resulting from regional demography, and the slow-down induced by the COVID-19 pandemic exacerbated it. As labour shortage increases, recurring recommendations state that companies should accelerate investments in robotics and automation, as well as reorganising their work to make it more sustainable for workers [2–4]. Moreover, according to Statistics Canada [5], total investment in Canadian building construction increased by 6.3% to reach $19.9 billion in April 2021, because of continued strength in the residential sector. This increase reflects rising demand and tightening supply for building materials since the start of the COVID-19 pandemic [5]. As a response to productivity issues and increasing demand in the construction field, off-site manufacturing, or modular construction, is of growing interest all over the world as a solution to the affordable housing crisis by means of providing housing supply more efficiently [6]. It also offers significant benefits over traditional on-site construction such as faster building erection, reduction of urban disturbances, reduction of material waste, better ergonomics for workers, better construction quality, and better control of the supply, inputs, and outputs. In Quebec and in North America in general, the most common construction method is the use of light-framed wood structures, which also prevails amongst off-site manufacturers. Light-framed wood structures have the particularity of being very light compared to steel or concrete buildings of similar volumes.

The lightness of the structures offers interesting advantages while adding complexity to the structural design due to the high tensile and shear forces induced by lateral loads [7].

Despite the fact that off-site construction shows benefits that are maximised when similar operations are repeated numerous times, as in the case of multistorey multiresidential buildings, off-site manufactured buildings other than single homes are uncommon in the actual industry of Quebec [8]. When asking professionals why this construction method is rarely chosen for multistorey multiresidential buildings, all answers led to the deficient assembly process which is, in its actual form, time-consuming [8–11]. The lack of an efficient assembly technique in modular construction is often cited as a problem in the literature, as well as being among the main disadvantages of off-site manufacturing [6,12–14].

### 1.2. Literature Review

As a response, many types of joints have been developed and tested in previous research. Sharafi et al. proposed an interlocking system to automatically enable connection and facilitate assembly with lower force and simpler motions [14]. Lacey et al. proposed simplified structural models for interlocking intermodule connection (IMC), allowing them to predict interlocking connection behaviour [15]. Chen et al. proposed new beam-to-beam tenant and bolt connections, which facilitate the horizontal connection of modules with a tenant acting as a horizontal geometrical constraint as well as vertically positioning the modules [16]. However, on-site installation of bolts is required in order to prevent disconnection in the case of a tensile load. Annan et al. studied a corner connection with plate and bolts located at the outer surface of the column junction, and analysed the structural behaviour [17]. Loss et al. proposed a new hybrid solution for modular steel–timber construction, including smart devices for screwing connection points [18]. Sendanayake et al. tested a variety of connection systems to assess the damping capacity of the connection assembly [19,20]. To dissipate energy through connection devices, Sendanayake et al. used the principles of ductility and damping capacity. The developed connector ensured the lateral connection of the modules via a complex assembly made of two metal plates, within which was included a resilient layer made from a material with a high damping coefficient. Dai et al. proposed a novel plug-in self-locking joint for modular steel construction which showed a high level of automation [21]. The proposed joint showed great potential for improving the efficiency of the assembly process in modular construction. Nonetheless, Dai's joint does not allow for disconnection in case of accidental connection nor for cases of waste management in the deconstruction process. Moreover, the design is suitable for steel construction but would require major changes if it were to be used in light-framed construction, since it was designed to be welded to a steel structure. To conclude, while many researchers have focused on the steel or concrete modular-building environment, the present literature review highlighted a research gap in light-framed modular construction, specifically with regard to the module-to-module connection and the overall load-bearing systems for discontinuous buildings (modular type).

### 1.3. Research Scope

Field observations were conducted in order to identify how a higher degree of automation in the assembly process could benefit off-site manufacturing, and hence reduce housing cost and on-site labour need. An automated assembly device was found to highly benefit off-site manufacturing and this study presents the development of a novel connecting device (NCD) in a multimodular prefabricated building context. The main purpose of this research was to develop an assembly solution that could improve the construction of multistorey modular buildings. This research contributes to extending the understanding of light-framed modular buildings as well as identifying how modular assembly could benefit from a new connecting device. This research also contributes to the advancements in this field by proposing a device that fulfils functional and structural requirements for light-framed modular connection.

## 2. Materials and Methods

### 2.1. Knowledge Gathering

Figure 1 presents the conceptual methodology scheme used in this work. The phase of knowledge gathering involved six different sources of information. First, the literature review was divided into three major components: existing connecting devices for other purposes (A), research on modular buildings and modular connecting devices (B), and light-frame construction methods (C). The literature review was completed by observations that were again divided into three major components: in-factory visits (D), on-site visits (E), and interviews with major actors (F). The in-factory visits (D) took place at Groupe Profab (Vallée-Jonction), Maisons Laprise (Montmagny), and Structures Ultratec (Saint-Apolinaire), all located in Canada. The on-site visits (E) took place in Quebec City at a construction site of a four-storey modular building including a total of 24 residential units. The project was led by a group of off-site manufacturers who joined their production capacities in order to make a multiresidential building out of modules. Further on-site observations took place in Vallée-Jonction at a six-plex modular construction site. Last, the interviews (F) were conducted with the CEO and structural engineer at Groupe Genius [8], the general manager and CEO at Maisons Laprise [9], the production manager and the CEO at Groupe Profab [10], the director of creation at STGM Architectes [11], and the director of engineering at Structures Ultratec [22].

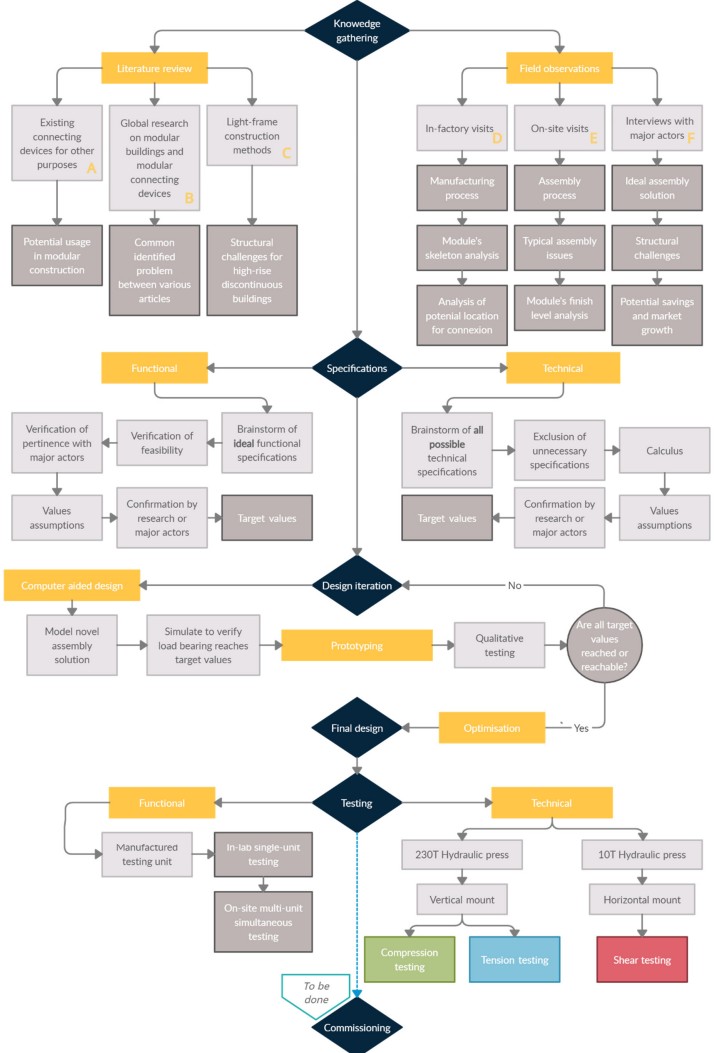

**Figure 1.** Specific methodology for the design of the novel connecting device.

## 2.2. Specifications

The problem definition was detailed through an exhaustive list of design specifications that acted as guidelines for the iterative design process. Design specifications were divided into two categories, functional specifications and technical specifications. All six sources of observations (A, B, C, D, E, F) identified in Figure 1 contributed to the development of the design specifications. For each specification, feasibility was verified with preliminary calculations. The specified preliminary values were confirmed or modified with major actors (F) until final target values were obtained.

Table 1 presents all the functional and technical specifications for the design of the novel connecting device (NCD), as required by the second step of the design methodology.

**Table 1.** Functional and technical design specifications for the NCD.

| Characteristics | Target Values | Comments | Sources |
|---|---|---|---|
| Movement of insertion | Vertical | In multistorey multimodular buildings, the modules are lifted by crane and their arrival on connecting site is always vertical. Hence, the parts of the NCD must insert vertically. | [9,10] |
| Locking mechanism | Automated, mechanical | The locking mechanism must replace the tensile bolts typically used to restrict pull-out motion without extra operation. | [8,11] |
| Module-mount in factory | Easy, optimal station for mounting | The NCD must be easily installed in an off-site factory at a specific workstation. The wood to metal linkage must be simple and efficient. | [9,10] |
| Module alignment | Easy, conic entry | Modules can have a length up to 60 feet. Corner alignment is ensured by workers on-site. Since a mechanical device often shows smaller dimensional tolerances than wood structures, a self-alignment design must ensure an easy connection. | [8,11] |
| Unlocking | Possible, less than 3 min operation | If, for any reason, the NCD must be disconnected, a less than 3 min operation must be designed. This feature is also essential for end-of-life considerations of a building. | [8,11] |
| Connection confirmation | Sound or visual | One must know if the connection was successful or not, in order to be confident in the structural stability of the building. Hence, the NCD must present a sound or visual signal of connection confirmation. | [8,11] |
| Minimum force required for connection | 450 N | To avoid undesirable triggering of the locking mechanism during transportation and installation, a minimum triggering force must exist. Hence, a 450 N opposition shall be sufficient to avoid an undesirable trigger. | [8,11] |

| Characteristics | Target Values | Comments | Sources |
|---|---|---|---|
| Internal access required | None | The main goal of this NCD is to eliminate all manipulations regarding the assembly of modules that would take place inside the building, allowing the manufacturers to complete the interior finish of the module. | [9,10] |
| Location | Inside the walls/floors | It is desirable to maintain the interior living space as large as possible. Hence, the connector must be hidden inside the walls or inside the floors, and avoid expanding inside the living space. | [8–10] |
| Range of usability | Standard range | The NCD can take a range of dimensions in order to fulfil the needed capacity for various needs. For example, the sixth floor of a building does not withstand forces as big as the first floor; different sizes of connecting devices shall exist. | [8] |
| Vertical load path | Unaffected | Load concentrations on corners must be avoided. Hence, NCD location and fixture must allow for a continuous contact between the header joist and the top plate of the two adjoining modules. | [8] |
| Lateral fixture | 1 corner 2 corners 4 corners | A modular building presents diaphragm discontinuity at every end of a module. In order to horizontally transmit the lateral loads from module to module, lateral fixtures at vertical connecting points must exist. It must account for the three most common geometries, namely a single exterior corner, two exterior corners, or four interior corners. | [11–22] |
| Tensile capacity | 200 kN | Lateral forces can induce a pivot-like behaviour of the building, and some of its modules will withstand tensile loads. | [8] |
| Compressive capacity | 1000 kN | Compressive forces might never reach such a high value if the header joist and top plate contact is preserved. However, in cases of beam failure, a critical load concentrated in corners must be withstood. | [8] |
| Shear capacity | 40 kN | Lateral forces will induce a staircase-like behaviour and the building's response will be a combination of the siding diaphragm and the ductility of the NCD. | [8] |

### 2.3. Design Iteration

During design iteration, different computer-aided designs were generated. Static simulations were run with SolidWorks Simulation to verify the load-bearing capabilities of the assembly. Parts were then 3D printed for functionality testing, and prototypes were manufactured for experimental testing.

The final design was then selected, and more complex calculations were carried out in order to optimise the design and ensure maximal performance, manufacturability, and cost, as well as loosening the tolerances needed.

Last, to ensure the final design was economically viable, an economic analysis was pursued and is presented in Section 3.3. The hypotheses underlining the analysis were based on field observations D, E and F.

*2.4. Testing*

The testing phase involved two distinct presses, set-ups, and protocols, the first one for shear testing, and the second one for compression and tension testing. The testing phase aimed to verify if the tensile, compressive, and shear capacities identified in the design specifications could be reached without plastic deformation. It also aimed to verify the results of finite element analysis (FEA) which would facilitate evaluation of the impact of future changes in the design. Hence, only two specimens were tested, one in compression and tensile loading (until tensile rupture), and one until rupture in shear loading. The results were used only for qualitative comparison between phases of force application. Table 2 presents the technical data of the specimens tested.

**Table 2.** Connector critical values, test-related.

| Characteristic | Value | Unit |
|---|:---:|:---:|
| Material (*for all parts except hardware*) | Aluminium 6061-T6 | *NA* |
| Material yield limit | 275 | MPa |
| Shaft outer ring diameter | 70.0 | mm |
| Shaft inner ring diameter | 53.5 | mm |
| Shaft length | 205.7 | mm |
| Shaft teeth thickness at pure shear region | 12.1 | mm |
| Frame height | 209.6 | mm |
| Frame width | 152.0 | mm |
| Frame length | 76.0 | mm |
| Frame conic entry smallest diameter | 80.6 | mm |

In order to analyse the behaviour of the connector itself, steel jigs were used and designed according to the typical environment of the connector, namely a corner header joist fixed to the connector via 12 bolts. The wood behaviour led to deformations of various types like hole ovalisation, perpendicular tearing, group tear, and more, which could affect the resultant force applied on the connector but were omitted in this experiment.

The equipment used for **shear testing** consisted of an 80 kN hydraulic press with T-shaped channels on the bed plate, allowing for strong fixation of the test jigs. The press was numerically controlled with a closed-loop feedback system that both regulated and tracked the position, force, and speed simultaneously. The position was controlled with a $\pm 0.01$ mm uncertainty and the force with a $\pm 500$ N uncertainty. With this set-up, the force was measured with a load cell and the displacements were measured with a position indicator located along the hydraulic jack; hence, the displacements obtained include the deformation of the hydraulic press itself and its components. Nonetheless, specific precautions were taken to maximise the accuracy of the results. First, with the shear testing set-up, since the loading protocol reached 94% of the hydraulic press nominal capacity, the main concern regarding undesirable displacements was a potential bending in the post holding the moving end of the hydraulic jack. If the post bent downwards because of the compression induced by the tension force in the hydraulic jack, the true displacement would be smaller than the one measured. To evaluate undesirable displacements, an indicator (distance amplifying instrument) was placed at this specific location throughout the entire test and maintained a displacement reading of 0 mm. Hence, the only possible machine deformation was within the hydraulic jack, which, from calculations, was expected to incur an elongation of 0.1128 mm when loaded with a 75 kN load. Moreover, the assembly set-up can undergo deformation that adds up to the deformation of the connector itself, but since the jigs were over-designed and very stiff, the set-up error was negligible. The sum of the set-up error value with the position reading device error value corresponded

to a total error estimated at 2.6% of the displacement obtained. Further static testing will take place prior to a set of dynamic experiments and the data will be compared with the ones obtained in the present experiment to ensure reliability of the data. The jigs illustrated in Figure 2a consisted of assorted 1020 cold rolled steel plates that were designed specifically to minimise deformation of the jig parts. Figure 2b illustrates a typical module corner in which the connector is located, and by comparing Figure 2a,b, the analogous configuration between the jigs and the real environment of the connector can be observed. Also note that the housing joint separation was positioned perpendicular to the shear load, which is weakest orientation of the mechanism. This design choice for the test bench had the benefit of recreating the worst-case scenario, leading to a more conservative security factor. Only one specimen was tested, and the maximal force applied was of 75 kN with a traction speed of 0.024 mm/s. The loading protocol for shear testing began at phase 1 with a progressive application of a 35 kN force and progressive withdrawal. The second phase consisted of the repetition of the first phase to ensure all functional gaps were eliminated during the first force application. This allows analysis of the assembly in a context where it underwent a repetitive loading case with the same force orientation and similar magnitude as a typical building. The last phase of the protocol was a 75 kN force progressive application and withdrawal. Through all phases, no opposite forces were induced to avoid completely reversed behaviour. The force–displacement graph associated with this protocol is presented in the results section.

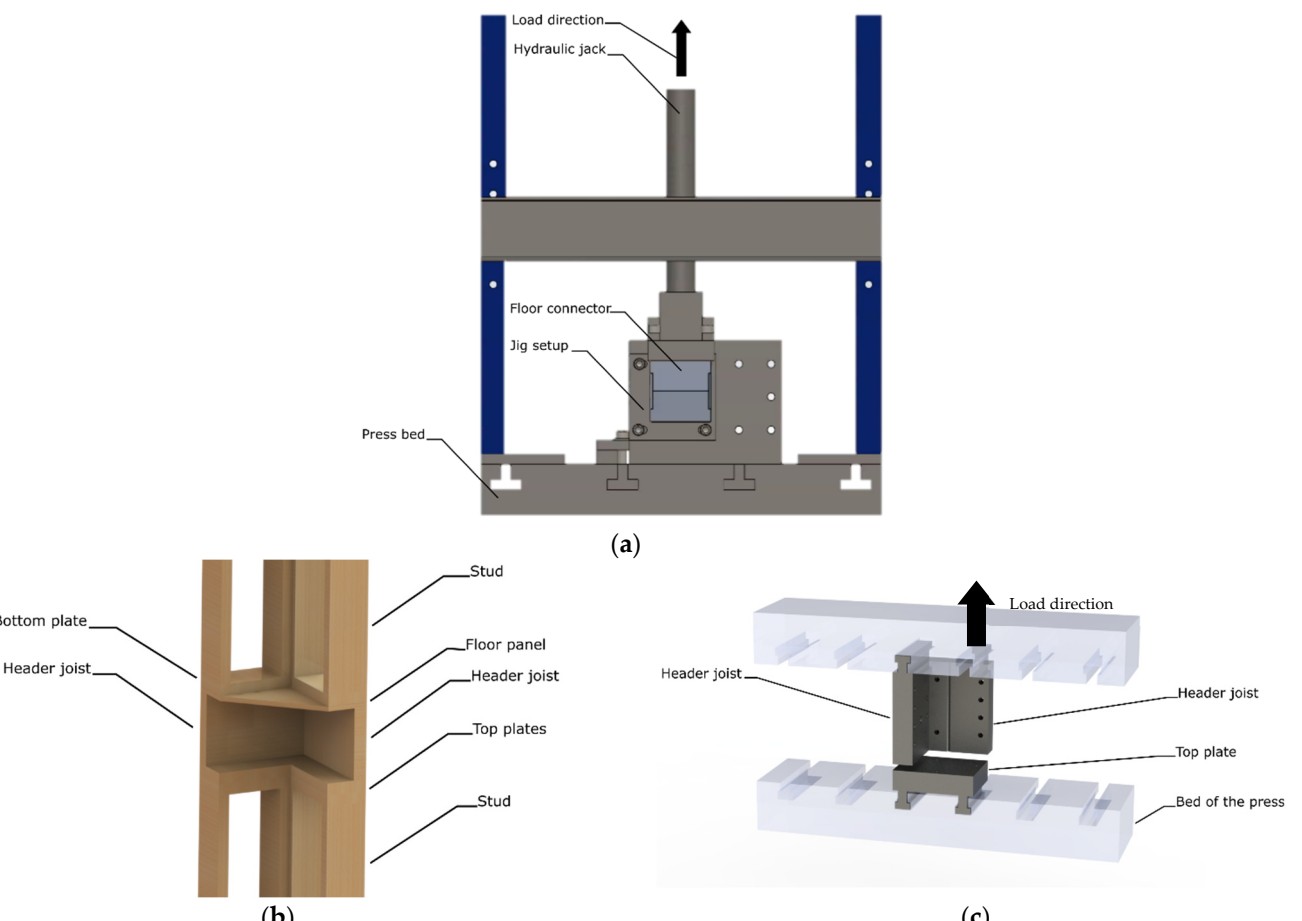

**Figure 2.** (**a**) 3D model of the jig mounted on the 10 ton hydraulic press for shear testing, (**b**) 3D model of a typical corner showing where the connector is located in the light-framed wood modules, (**c**) 3D model of the jig mounted on the 230 ton hydraulic press.

The equipment used for **tension and compression testing** consisted of a 2000 kN hydraulic press with T-shaped channels on the bed plate, allowing for a strong fixation of the jigs to the press. The press was numerically controlled with a closed-loop feedback system that regulated and tracked the position, force, and speed simultaneously. The uncertainty as pertains to each variable was as follows: $\pm 0.005$ mm in position and $\pm 3.5$ kN in force. In this specific set-up, the displacement was measured by the change in distance between the two rigid beds that acted as holders for the assembly. Hence, if parts of the hydraulic press incurred deformation throughout the test, it did not influence the distance between the beds, and consequently no error was induced in the position reading. In between the bed plates, the assembly set-up could undergo deformation that adds up to the deformation of the connector itself, but since the jigs were over-designed and very stiff, the set-up error was negligible. The jigs are illustrated in Figure 2c and consisted of assorted 1020 cold rolled steel holders that were designed to be analogous to the real connector environment illustrated in Figure 2b. The jigs were fixed to the connector with the same bolts as would be the header joist. The tension testing loading protocol began with the progressive application of a 116 kN force. The second phase consisted of the withdrawal of the force until a 42 kN load remained on the assembly. The third phase was the progressive application of a 200 kN force. The fourth phase was a progressive withdrawal until a 51 kN load remained. The fifth phase was the progressive application of a 238 kN force. The last phase was a progressive withdrawal until no force remained on the assembly. Again, no opposite forces were induced throughout the protocol to avoid completely reversed behaviour. Finally, a last force application was applied until rupture to identify the failure mode. The compression testing loading protocol consisted of a progressive 800 kN application and withdrawal (first phase) and a progressive 1200 kN application and withdrawal (second phase). The force–displacement graph associated with the tensile protocol as well as the graph associated with the compressive protocol are presented in the results section.

## 3. Results

### 3.1. Description of the Problem

Field observations were performed in three different factories and at a construction site in Quebec City where a four-storey, 24-unit multiresidential modular building was under construction. These four sources of knowledge gathering are detailed in Section 3 and led to the following observations.

Currently, during modular building assembly, every existing connection relies on a multistep process. When the upper module is completely laid down on the lower module, workers must enter the building at various connection access points and fix screws and bolts to ensure a permanent link between the modules. This operation is time-consuming. It has been observed that workers can easily spend an hour securing the connections before another module can be assembled. Since buildings are subjected to different types of loading (compressive, tensile, shear) over the course of their existence, connecting points must restrict any displacement caused by external loadings, which explains the need to add axial fixtures on the vertical load paths. For corner-supported modules, all four corners in each module have to be left unfinished upon delivery (e.g., drywall not installed, plaster joints not done, and paint not applied) to allow working space to install additional fixtures. In the case of a longer module, connection points in need of additional work can number up to eight. This leads to a high percentage of work left to be completed on-site rather than in the factory, consequently reducing the benefits of off-site manufacturing.

Once internal connections are completed, further steps are required at the exterior surfaces. Depending on climate, usage, and materials, the vertical and horizontal edges of adjoining modules have to be finished to integrate isolation, diaphragm continuity, air barrier, vapour barrier, sound barrier, and siding. In Quebec, most off-site manufacturers create light-framed wood modules [9,10,22,23]. Typical modules have to be freestanding and show a high level of rigidity for transportation and handling needs. From bottom

to top, the floor is generally made of an engineered-wood perimeter acting as header joists. They contain the floor joists and allow for electricity and plumbing services. This assembly is topped with wood panels, for which the structural role is to act as a diaphragm, controlling shear in the floor plane. Typical walls are mounted on top of the wood panels, beginning with bottom plate and studs, up to the top plate. Again, vertical wood panels are fixed at the outer surfaces of the walls to oppose shear forces. Insulation, plumbing, and electricity are housed between studs, some sheeting is added to act as a vapour barrier, and the drywall is finally installed. The external surface of the top headers either becomes the sitting surface for the roof joists and roof ruts, or the sitting surface for the next floor header joists. In multistorey cases, the common surface between the top header and the header joist is where connection has to take place in order to transfer loads between stacked-up modules.

For security purposes during building assembly, for every module added, all of the connecting steps mentioned above have to be completed before lifting the next module. Indeed, if load-resisting systems are not deployed at each module addition, the building is subject to collapse at any moment. This induces wait times for many workers on-site which reduces the overall efficiency. Multistorey buildings that comprise numerous modules may be manufactured by a combination of independent manufacturers because of limited storage and production capacities. Since no standard connecting device exists, manufacturers must agree on a connection technique and modify their production lines accordingly. Since this often differs from their regular technique, it becomes a potential source of errors and complications. The cost and delays induced by this design modification phase are of great importance and often discourage promoters from choosing this construction method. This highlights the importance of designing a standard connecting device that can easily be implemented in light-framed modules regardless of the manufacturer.

*3.2. Proposed Novel Connecting Device*

The proposed NCD is composed of a lateral plate (LP) and two distinct assemblies, respectively named floor connector (FC) and ceiling connector (CC), presented in Figure 3a. Figure 3b presents the CC in an exploded state, and Figure 3c presents the FC in an exploded state with local enlarged views of small subassemblies. Figure 3d presents a cross section of the assembly prior to connection, and Figure 3e a cross section of the assembly when fully connected.

The CC is composed of a ceiling plate (1) which ties down onto the top wall plates. The ceiling plate (1) has a central cavity into which a serrated shaft (2) is inserted. The FC assembly is made of a frame (3) fixed to the floor beams. The frame (3) has in its centre a circular opening slightly greater than the diameter of the shaft (2), with which is aligned a trigger block (4) of cylindrical shape, almost entirely hollowed out except for a lower ring, columns on the periphery and a solid section at the top. This trigger block (4), when in the initial position (low position), geometrically constrains (by its diameter) the four clamps (5) away from the centre of the assembly. The clamps (5) have a geometric profile complementary to that of the shaft (2), and are linked to the frame (3) via springs (6) in compression which tend to push the clamps (5) towards the centre. When the assembly begins, the shaft (2) progresses in the cavity of the frame (3) until it reaches the upper section of the trigger block (4). The trigger block (4) is then driven in movement upwards until both cylindrical parts of the trigger block (4) reach the two cavities in the clamps (5) intended to accommodate the lower ring and the upper section of the trigger block (4). The clamps (5) are then geometrically free to move towards the centre to interlock with the shaft (2) and to create a final connection. The assembly is then completed, and the FC cannot be separated from the CC by tension, nor by compression, nor by shear. The frame plates (7) complete the assembly and close the cavities for two of the four clamps (5). The frames (2) and the frame plates (7) have four clamp access holes (8) used to reach the clamps (5). When screwing a safety bolt into a clamp access hole (8), the head of the bolt is pushed onto the frame (3) while the threads of the bolt apply a pulling motion on the clamp (5).

This eliminates the contact between the shaft (2) and the clamp (5) to allow unlocking for end of building life or to undo an unwanted connection.

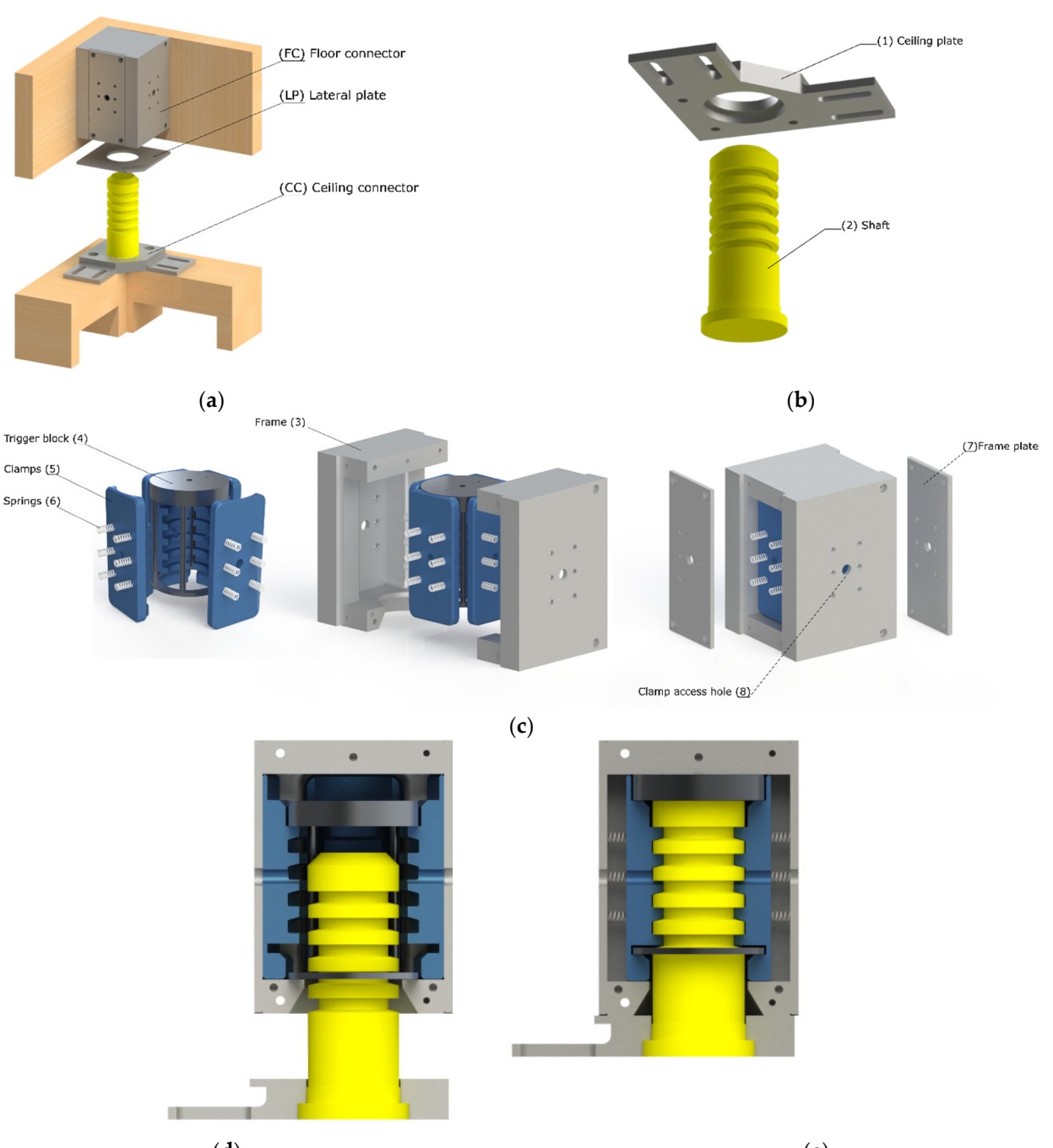

**Figure 3.** (**a**) 3D model of the connector in a wood module, (**b**) Exploded view of a 3D model of the ceiling connector (CC), (**c**) Exploded view of a 3D model of the floor connector (FC), (**d**) Section view of a 3D model of the connector prior to connection, (**e**) Section view of a 3D model of the connector after connection.

While the CC and FC ensure a vertical permanent linkage between modules on two separate floors, linkage between adjacent modules also has to be ensured. Hence, extra space was designed for an additional part that would allow horizontal linkage between modules, namely LP. Sendanayake [19] showed that the composition of this lateral part for linkage must be well-designed and optimised to ensure compliant behaviour and to

smoothly transfer shear loads between adjacent modules. Hence, the detail of the plate still has to undergo a complete analysis and rigorous design process, and will be covered in future research. With regards to the assembly process of this part, it was designed to be manually added directly on-site after each module addition, prior to installing the next storey. The plate is designed to require no fastening operations, since its positioning will be secured by the weight of the upper modules. Hence, the workers will only have to place the plates.

### 3.3. Potential Productivity Gain

To assess the economic advantages of using an automated connecting device, a comparison was established between the costs of erecting a multistorey modular building with and without the NCD. Current practice for single-home modular buildings allows for an off-site finish level of approximately 90% [24]. For multistorey modular buildings, the actual obtainable off-site finish level without NCD is approximately 60%, since floor panels and drywalls have to be left opened at connecting points (four to eight per module) [9,10,22].

In order to analyse the losses induced by these on-site finishing steps, conventional construction and off-site construction were compared. Several studies have indicated that modular construction can reduce the construction period by 50–60% compared to the traditional method [6,23]. From field observations, it can be estimated that the time needed to do the same job on-site will take twice the time needed to do it in the factory, attributable to off-site organisation and easy material supply [25,26]. Knowing that rates are, at the time of writing, 65$/h for traditional construction and 32$/h for off-site construction in Quebec, a one-hour job will cost 32$ to do off-site, and 130$ to do on-site [9,10]. This corresponds to a 400% cost increase for any specific job. This highlights the need for a NCD that maximises the off-site completion of modules.

The following economic scenario illustrates the potential productivity gain of the automated NCD. A 500 square foot module can present a maximal off-site finish level of 90% with a total off-site labour cost of 20,000$ [10]. Hence, off-site, the completion of the module, labour-wise, costs 222$ for each 1% completed. Following the 400% cost increase, this means 1% module completion labour-wise costs 888$ on-site, compared to 222$ off-site. Table 3 and Figure 4 below illustrates the linear effect of these cost variations. Of course, this linear effect could show exponential behaviour in highly repetitive buildings, since workers show a greater efficiency when repeating the same operations.

**Table 3.** Hypothesis for economic scenario and graph.

| Characteristic | Value | Source |
|---|---|---|
| Typical total cost per 500 ft$^2$, 90% finish module | 75,000$ | |
| Typical labour cost per 500 ft$^2$, 90% finish module | 20,000$ | |
| Typical fixed cost per 500 ft$^2$ (material, factory costs, transport, etc.) | 46,111$ | [9] |
| Cost for 1% module labour in-factory | 222$ | |
| Cost for 1% module labour on-site | 888$ | |

Figure 4 presents the total cost per 1% of a 500 ft$^2$ module as function of its finish level upon delivery. The total cost includes the typical fixed-costs of 46,111$ per module (independently of the finish level since materials and transport represent most of the fixed-costs), the off-site labour costs, and the on-site labour costs. This highlights that a 90% completed module presents a typical total cost of 75,000$, whereas a 60% completed module presents a typical total cost of 95,000$. The balance of the plant is not included, and nor are the assembly costs, the connection to municipal services, excavation, or any other cost unrelated to the modular manufacturing.

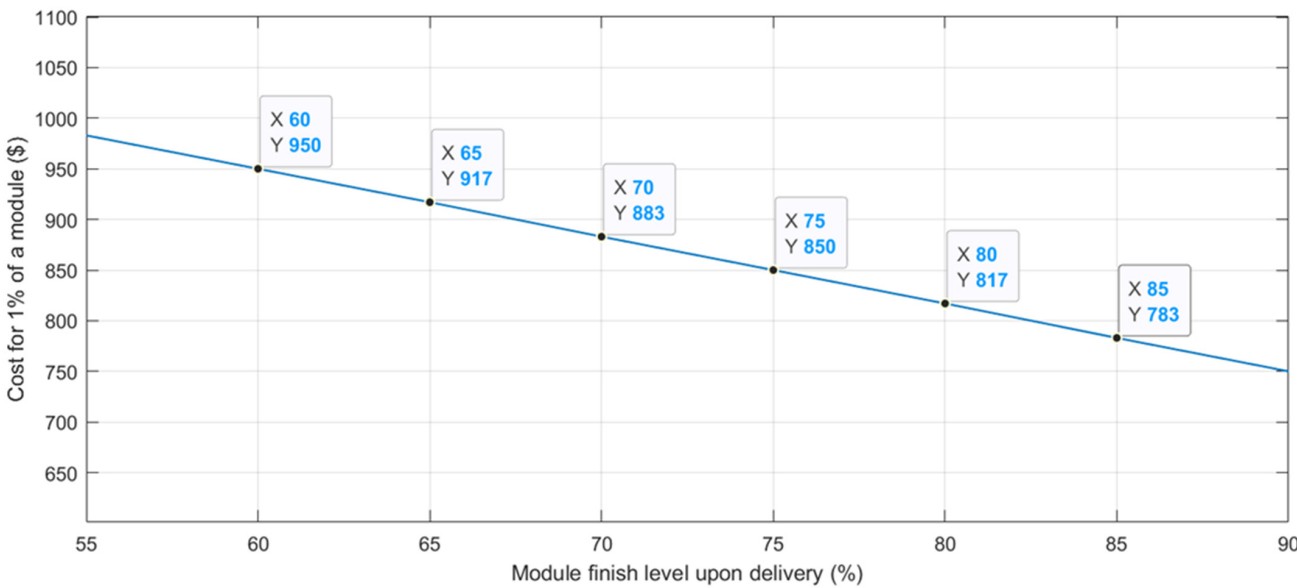

**Figure 4.** Cost for 1/100 of a module as a function of its finish level upon delivery.

Through the development of this NCD, a 20% increase in completion level upon delivery is expected, which corresponds to an equivalent of 13,333$ of savings on the labour cost for every module according to our hypothesis. Many other additional costs are expected when delivering modules at a low finish level, such as supply of material and tools on-site, labour management, risk management, and more. The costs of the connectors themselves are not included, but are expected to be offset by the structural changes induced by their use, such as the elimination of nails, metallic plates, and bands, for example.

It is also of interest to analyse the savings associated to a more efficient assembly process on-site. A typical 8 h day of assembly involves ≈15 workers (65$/h), one crane (≈6,000$/day), and ≈four truckers (≈45$/h); thus, every assembly day costs 15,240$. Without the NCD, one module is assembled every hour, corresponding to 1905$ per module assembled. With the NCD, it is expected that tasks can be done simultaneously and the connection does not involve any manipulation, thus reducing the assembly time to 15 min per module, corresponding to 476$ per module assembled. This allows a 75% cost reduction at assembly day, corresponding to approximately 1,429$ of savings per module.

Combining the savings of the labour cost and the savings of the assembly day leads to a total cost reduction of 14,700$ per module when using the NCD.

### 3.4. Testing

To ensure that the NCD fulfils the technical requirements, static monocyclic testing in two configurations was used to test the compressive, tensile, and shear capacities.

The energy dissipation capacity and the equivalent stiffness of the assembly were of great interest for characterising the structural behaviour of the connector. In most regulating building codes and norms, accessories are qualified according to their stiffness and, as depicted in many articles, connecting devices must show damping capacity [13,27–30]. The following section first presents a general analysis, followed by a stiffness analysis and an energy dissipation analysis.

### 3.4.1. General Analysis

Figure 5a illustrates the displacement induced by the lateral loads that create a separation plane in the NCD. When a lateral force is applied on the FC, a horizontal gap is created between the FC and the CC. This gap creates a loading case which will be referred as shear loading. The two major points of interest in this loading case are the bearing stress at the interface between the shaft and the frame (as can be seen in detail A of Figure 5a) and the

thread pullout at the junction of the two frames (as can be seen in detail B of Figure 5a). Since the frame is made of aluminium and the bolts are made of steel, with this specific dimensioning, thread pullout is more likely to occur than any type of breakage in the bolts. To oppose the joint separation between the two frames, four $\frac{1}{4}$-20 bolts were used to link the two frames together (only two are visible in the section plane of Figure 5a) and four $\frac{1}{2}$-13 bolts were used to link the right frame to the header joist (not visible in the section plane). Figure 5b presents the deformation of the shaft due to the force exerted at the point of contact with the frame (with a deformation scale of 400:1).

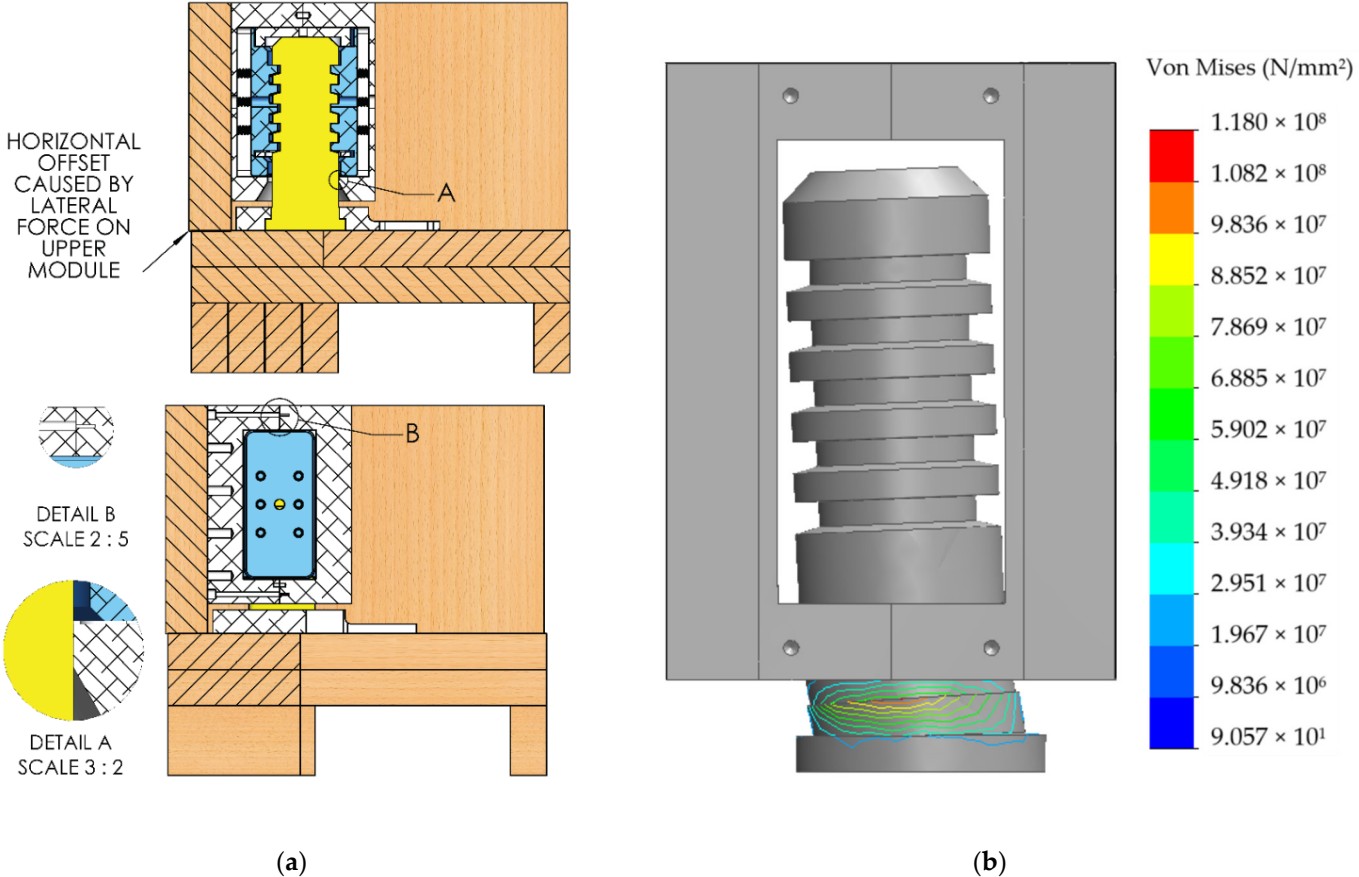

**(a)**          **(b)**

**Figure 5.** (**a**) 3D model of the physical motion created by shear loading, (**b**) FEA simulated deformation of the shaft caused by shear loading (deformation scale 400:1).

Figure 6 presents the force–displacement graph showing all three phases of force application in shear. The first two phases did not show any major plastic deformation, while the third phase showed a bolt thread pullout at the joining area between the two frames induced by a 59.6 kN force. The observed bolt thread pullout highlights that this specific loading configuration, namely lateral loading originating from inside the module towards the header joist, can lead to a sudden rupture. Prior to the shear testing, finite element analysis showed that a 30 kN lateral force applied at the base of the FC would create a 275 MPa stress at the weakest point of the assembly, which corresponded to the elastic limit of the material. This weakest point is a high-stress-bearing area shown as point A in Figure 5. After all three phases, no visible permanent deformation was observed at the bearing surface contact, which confirms that bearing stresses can be neglected if the permanent deformation is small enough to have no effects on the functionality and strength of the assembly.

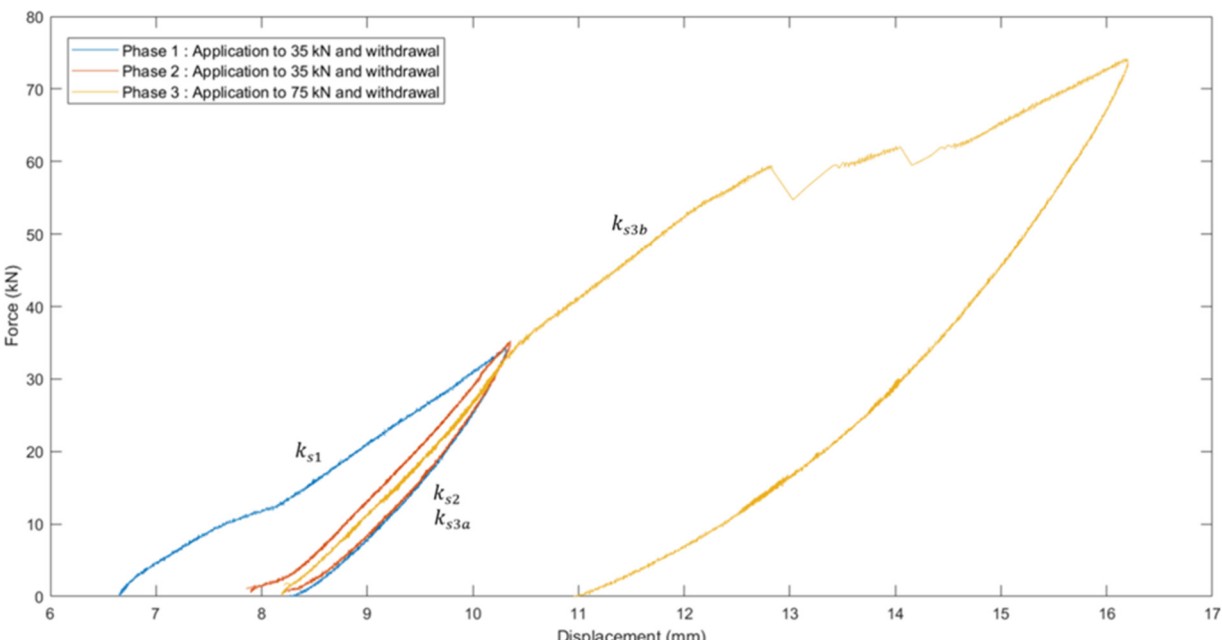

**Figure 6.** Shear loading, all phases of force application.

Figure 7 presents the force–displacement graph showing all six phases of force application in tension. No sign of permanent damage was observed throughout this protocol. Moreover, the constant linear behaviour in application phases 1, 3, and 5 shows the absence of permanent deformations. To conclude the tension testing, a tensile force was applied until rupture, which occurred at 300 kN. The shoulder of the shaft presented a pure shear failure mode.

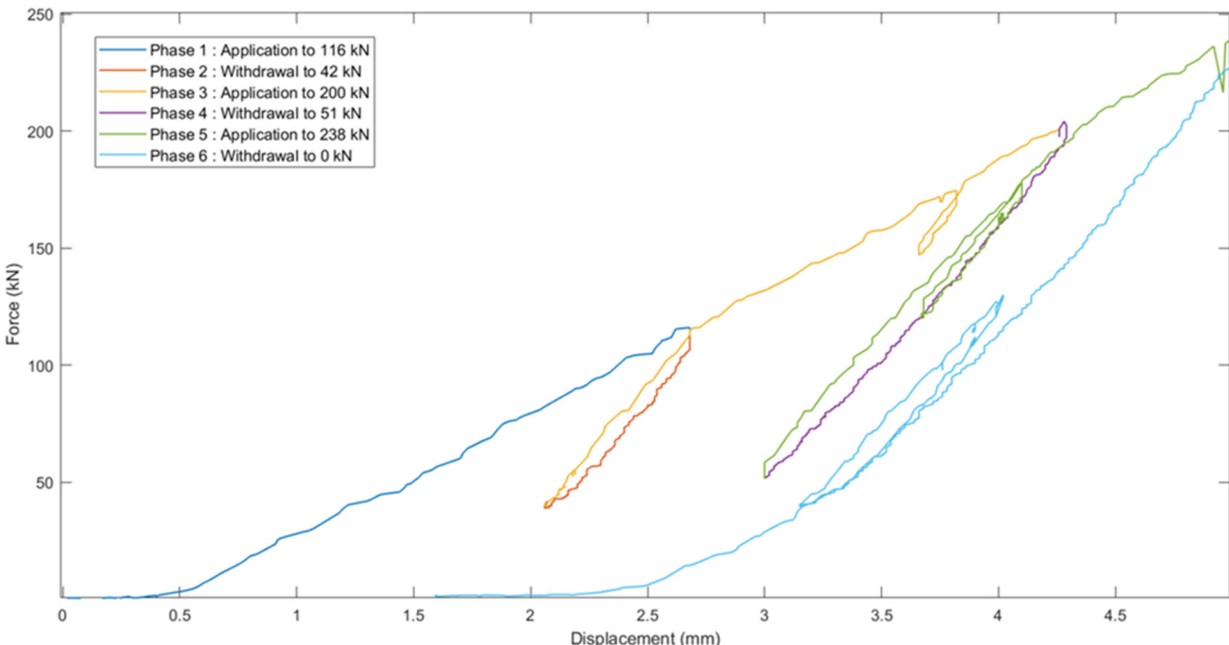

**Figure 7.** Tension loading, all phases of force application.

For the compression testing, the results are presented in Figure 8. The withdrawal data were omitted from the graphs since they were highly noisy and hardly interpretable. This was due to the type of command used in this protocol, which differed from the type of command used for shear testing. In this case, the force application was controlled with a

steady speed of application which generated a fixed number of values at every second. The withdrawal of the force followed a unit step command which substantially reduced the number of values obtained. Moreover, the hydraulic system was more stable in pressure increase than in pressure decrease, inducing hysteresis effects caused in part by friction in the seals. The curves obtained from the force application data displayed very similar slopes in both phases of force application. From visual observations, no permanent deformations were noted throughout the compressive testing. FEA calculations showed that no plastic deformation should occur under 1200 kN.

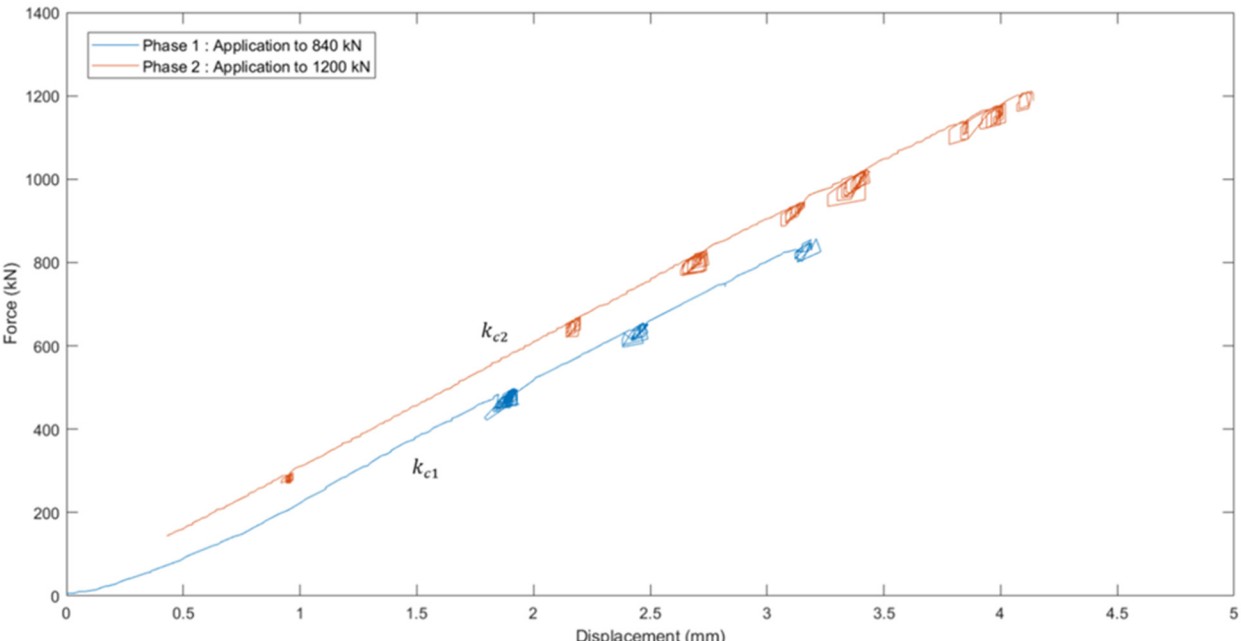

**Figure 8.** Compression loading, all phases of force application (no withdrawal).

3.4.2. Stiffness Analysis

Since the stiffness of the connector highly depends on the initial position of the internal parts of the assembly and previous force applications, stiffness will be noted as $k_{ij}$. The I suffix stands for the type of loading, with s for shear, c for compression, and t for tension, and the j suffix refers to the phase of load application (referring to Figures 6–8). Stiffness was calculated using the linearly constant parts of the data, and was calculated as follows [30]:

$$k_{ij} = \frac{\Delta F_{ij}}{\Delta x_{ij}}$$

Figure 6 shows that for shear testing, the first phase (35 kN force application) led to a displacement of 1.605 mm while the second phase (35 kN force reapplication) led to a displacement of 0.310 mm. The stiffness values obtained were $k_{s1}$ = 21.81 kN/mm associated with the first force application, $k_{s2}$ = 112.90 kN/mm associated with the force reapplication, and a combination of $k_{s1}$ and $k_{s2}$ in a bi-linear form for the third loading phase, as follows:

$$k_{s3} = \left\{ \begin{array}{l} k_{s1}, \ y = [35, 70] \\ k_{s2}, \ y = [0, 35] \end{array} \right.$$

Figure 7 shows that for tension testing, when the force application reached a value for the first time, the associated slope was very similar for phases 1, 3 (the second part of the bi-linear slope), and 5 (the second part of the bi-linear slope). The same phenomenon was observable with the withdrawal phases 2, 4, and 6, which again showed very similar slopes. Phases 3 and 5 showed a combination of a force reapplication and a newly

applied force, which led to bi-linear slopes. Indeed, the force reapplication created a hysteresis phenomenon since the internal state of the assembly differed from the initial state. Subsequently, once the reapplied force was equal to the force applied before withdrawal, the slope changed to meet the previous slope. The global force application of 238 kN led to a displacement of 4.43 mm. This led to a stiffness $k_{t1}$, valid for the first application of force that ranged between 0 and 238 kN. The reapplication phase 3 increased the force applied from 42 kN to 116 kN, impacting the displacement from 2.07 to 2.71 mm. This led to stiffness $k_{t3}$, valid for a force reapplication or withdrawal within this range. The reapplication phase 5 increased the force applied from 53 kN to 206 kN, impacting the displacement from 3.00 to 4.37 mm. This led to stiffness $k_{t5}$, valid for a force reapplication or withdrawal within this range. The stiffness values obtained were $k_{t1} = 53$ kN/mm, $k_{t3} = 116$ kN/mm, and $k_{t5} = 112$ kN/mm. Figure 9 shows the superposition of the approximated curves for stiffness identification for all phases of force application in tension loading.

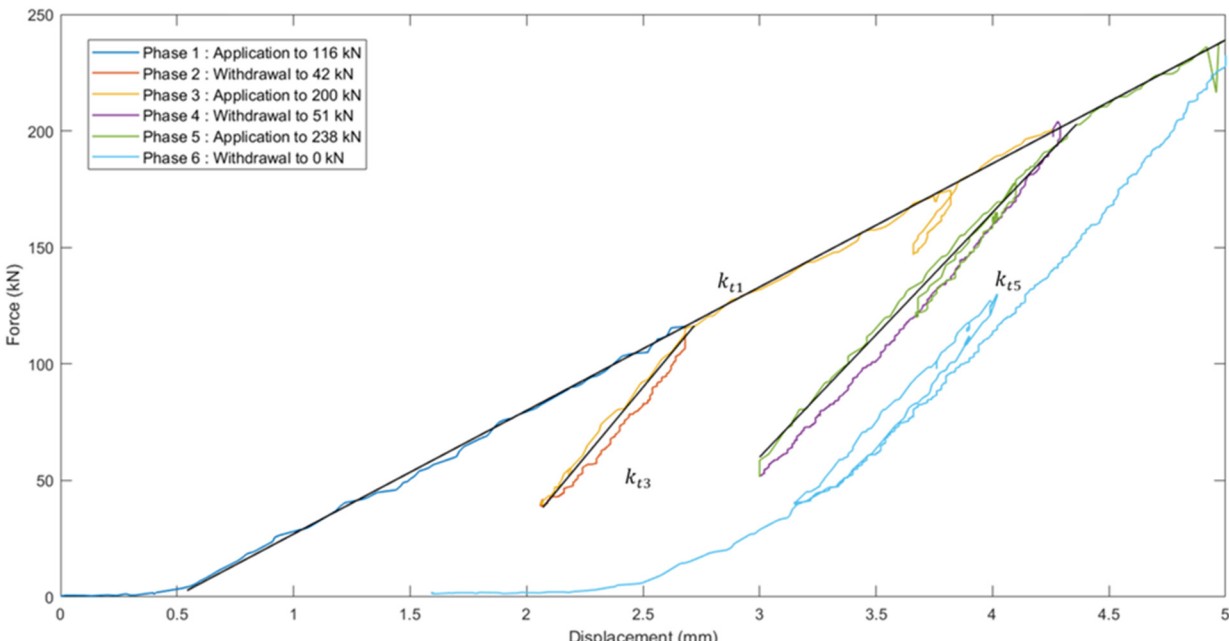

**Figure 9.** Tension loading, all phases of force application, with approximated curves for stiffness identification.

Figure 8 shows that for compressive testing, the global force application of 800 kN led to a displacement of 2.99 mm, followed by a 1206 kN force associated with a 4.11 mm displacement. The stiffness values obtained with approximated curves (noise cancelling) were $k_{c1} = 287$ kN/mm and $k_{c2} = 297$ kN/mm.

### 3.4.3. Energy Analysis

The comparison of phases 1 and 2 of shear testing was used to evaluate the difference in energy absorption between an initial force application and a force reapplication. The energy dissipated during a loading case will differ depending if the previous loading condition had the same direction or not. A repeated force in the same direction will affect the motion of the internal parts, since the gaps that allowed energy dissipation became filled. Figure 10 shows the integrated area which corresponds to the dissipated energy in the first phase (Figure 10a) and the second phase (Figure 10b).

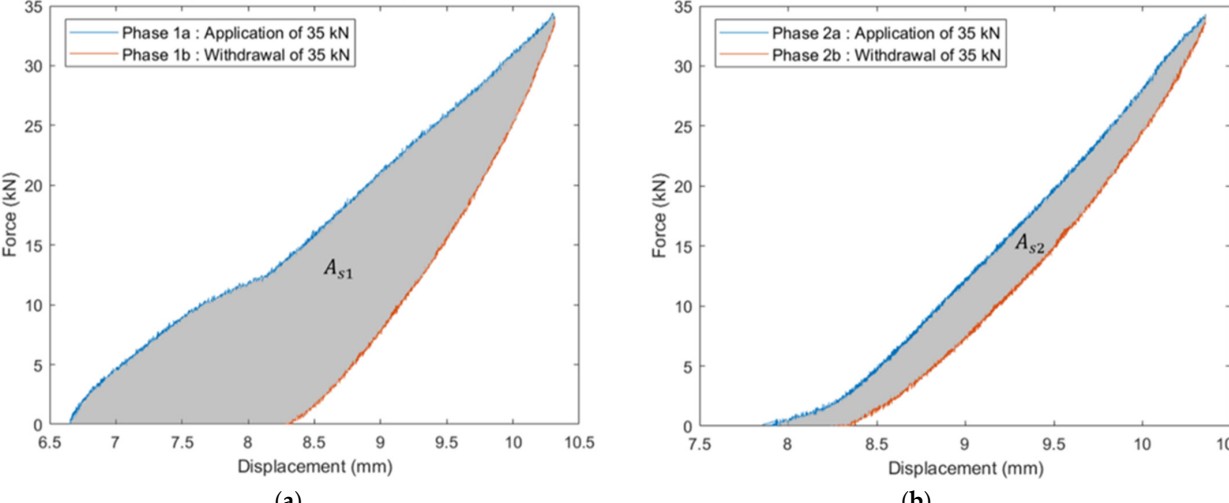

(a)

(b)

**Figure 10.** (**a**) Shear loading phase 1, dissipated energy area shown in grey (As1), (**b**) shear loading phase 2, dissipated area shown in grey (As2).

This dissipation of energy would allow, in a structural context, mitigation of the horizontal load transfers, acting like a damping device [31]. When not subjected to completely reversed loads, subsequent load application will induce a reduction of the dissipated-to-total energy ratio and an increase in the stocked-to-total energy ratio. Since energy absorption is valuable in a structural context, dissipated-to-total energy ratios should be established and compared. Total energy combines the dissipated energy and the elastic energy, and is obtainable with the integration of the force application curve. Hence, the dissipated-to-total energy ratio was 55.5 for the phase 1 in shear and 23.4 for the phase 2 in shear.

The tension loading was analysed exclusively with the total load application and withdrawal of 238 kN. The dissipated energy area was contained between the k_t1 curve and the 6th phase curve presented in Figure 9. Again, the dissipated-to-total energy ratio was of major interest in this context of initial force application in this specific direction, and showed a values of 53.7, very similar to the initial shear application ratio.

The compression loading led to noisy results regarding the force withdrawal, again cause by the type of command value and the hydraulic instability in the pressure decrease. Hence, an approximated curve was developed, leading to a dissipated-to-total energy ratio of 10.8. This much lower value was expected, however, since critical areas under compressive loading are much larger than in shear or tension, and hence the assembly presented a much higher stiffness and lower damping capacity when subjected to compressive loadings. Table 4 summarises all the results obtained through testing.

**Table 4.** Stiffness, energy values, and energy ratios obtained from calculations.

| Loading | $k$<br>[kN/mm] | $E_{total}$<br>[J] | $E_{dis}$<br>[J] | $E_{elas}$<br>[J] | $\frac{E_{dis}}{E_{total}}$<br>[%] |
|---|---|---|---|---|---|
| Shear phase 1 | 21.81 | 61.68 | 34.26 | 27.42 | 55.54 |
| Shear phase 2 | 112.90 | 37.26 | 8.73 | 28.54 | 23.42 |
| Shear phase 3 | Bi-linear | 365.48 | 222.97 | 142.51 | 61.01 |
| Tension (238 kN) | 52.32 | 548.80 | 289.71 | 259.09 | 53.78 |
| Compression phase 2 | 297.00 | 3092.60 | 332.05 | 2760.50 | 10.75 |

## 4. Conclusions

In this study, the development of a complete connection solution for modular light-framed wood buildings is detailed. The methodology behind the knowledge gathering

comprised a literature review of existing connecting devices as well as factory visits, on-site observations, and interviews with major actors. The combined knowledge led to the development of requirements under two distinct categories of functional and technical design specifications. Design iteration, final design, and testing are also detailed in this article. The main results of this study were as follows.

On-site observations highlighted major efficiency issues. First, the multistep assembly process is inefficient since every module must be permanently linked before pursuing the addition of modules. Additionally, modules have to be delivered at a poor finish level to allow access for connection on-site. Regarding the design phase, it is currently slow because of the absence of standard connecting devices.

The design specifications were detailed as follows. The movement of insertion must be vertical, the locking mechanism must be automated, the module mounting must be easy in the factory process, and the connecting device must help the module alignment. Furthermore, unlocking of the connecting device must be possible and take less than 3 min, the connection must be confirmed via sound or visual proof, the force required for connection must be at least 450 N, and no internal access must be required for connection. Additionally, the connecting device must be located inside the walls and floors to allow for module completion, the connecting device must be usable in the standard range, and the vertical-compressive load paths must be unaffected by the connector. Last, the tensile capacity must be at least 200 kN, while the compressive capacity must be at least 1000 kN and the shear capacity must be at least 40 kN.

The proposed connecting device has two parts (FC and CC), and meets all the design specifications. The connector's parts are vertically inserted directly into one another on-site and internal parts undergo displacement that triggers the locking mechanism. To ensure easy mounting in the factory, the FC is installed in the floor station simultaneously with the perimeter creation of header joists. Upon installation, the safety bolts are removed and the connector is charged, ready to clamp. The CC is installed in the wall station once the walls are up. For alignment purposes, the shafts can be easily located by the crane operator for module descent, and the shaft top face presents a significant chamfer which tends to centre the shaft in the conic hole located at the bottom of the floor connector, and hence facilitates the alignment process. For disassembly, the pulling out of the four clamps take less than 3 min. For connection confirmation, the second the trigger block is out of the way, the clamps are pushed to the shaft and the contact of these metallic surfaces makes a powerful sound. On assembly day, each worker assigned to a corner will hear the connecting sound and will confirm the connection. To ensure an opposition force at connection, the four clamps, when in the initial position, apply pressure towards the centre because of the compressed springs. This pressure is applied on the trigger block, which creates a friction force that opposes to the movement of the trigger block. To oppose this friction force, 450 N is needed. To maximise internal in-factory finish, no internal access is required and the connector is completely hidden inside the floor system. To ensure a good range of usability, the designed connecting device can be fabricated bigger or smaller depending on the load-bearing capacity needed. Finally, the designed geometry allows for continuous contact between the header joist and the top plate, hence not concentrating the load on the corners and not affecting the vertical load paths.

The economic analysis showed that if the NCD allows modules to be delivered at a 20% higher level of completion (from 60% to 80%), this will corresponds to approximately 14,700$ worth of savings per module.

The testing results showed that for the prescribed capacities, 1000 kN in compression did not induce any plastic deformation and nor did 200 kN in tension, and 40 kN in shear induced permanent deformation at bearing contact but no significant damage and no change in functionality. It also corresponded to approximatively 5/8 of the force that induced thread pullout.

The stiffness analysis showed that the stiffness of the assembly highly differs depending on whether the load is being applied in a direction different to the previous one or if

the load is being applied in the same direction as the previous load. The displacement of internal parts led to a different behaviour regarding the amount of energy being stocked or dissipated and changed the rigidity of the assembly.

Further testing is required to complete the characterisation of the NCD. Dynamic loading protocols should allow testing for fatigue and seismic behaviour. Moreover, thermal analysis should be done to ensure that the dew point is located outside the structural members, since wood does not maintain its properties when subjected to a higher humidity level. Future works should also analyse the complex behaviour of a light-framed, modular multistorey building to evaluate whether the design is suitable or not. Following the suggested methodology, the next design step will involve full-scale on-site experiments.

**Author Contributions:** Conceptualisation, L.P., A.B.-D. and P.B.; methodology, L.P., A.B.-D. and P.B.; validation, L.P., A.B.-D. and P.B.; formal analysis, L.P.; investigation, L.P.; data curation, L.P.; writing—original draft preparation, L.P.; writing—review and editing, L.P., A.B.-D. and P.B.; visualisation, L.P.; supervision, A.B.-D. and P.B.; project administration, L.P., A.B.-D. and P.B.; funding acquisition, P.B. All authors have read and agreed to the published version of the manuscript.

**Funding:** This research was funded by Natural Sciences and Engineering Research Council of Canada, through its IRC and CRD programs (IRCPJ 461745-18 and RDCPJ 514294-17), as well as the industrial partners of the NSERC industrial chair on eco-responsible wood construction (CIRCERB), the industrial partners of the industrialized construction initiative (ICI) and the Créneau Accord Bois Chaudière-Appalaches (BOCA). It was also funded by NSERC RGPIN 2015-04564.

**Conflicts of Interest:** The authors declare no conflict of interest. The funders had no role in the design of the study; in the collection, analyses or interpretation of data; in the writing of the manuscript; or in the decision to publish the results.

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
