# Peer review of "Assembly Solution for Modular Buildings: Development of an Automated Connecting Device for Light-Framed Structures"

_buildings, doi:10.3390/buildings12050672_

Round 1

Reviewer 1 Report

  • In the abstract section, there are too many descriptions about the background and lack of results and conclusions of current paper, please modify and supplement.
  • In the introduction section, the authors summarize some existing connections between modular structures, including self-locking connections. What are the problems with existing connections that requires new joints to be developed?
  • In the materials and methods section line 119, the authors state that“The testing phase occurred with two distinct setups and protocols, the first one for shear testing, and the second one for compression and tension testing.”and“No external sensors or devices were used to evaluate the displacements induced, which lead to results having little accuracy.”the authors are suggested to explain how to ensure the reliability of the data?
  • Figure 2 shows models of the jigs used in the tests. What is the overall loading device for the test? Please add corresponding schematic to illustrate, which can help readers to understand this study.
  • Chapter 3 aims to discuss the results of the test, section 3.2 and the material properties and dimensions of the specimens used in the test should be mentioned before this chapter.
  • In section 3.3, the composition and assembling method of NCD are described. How is the lateral plate assembled? What is its role in the connection?
  • Figure 5 can’t clearly represent the deformation and failure caused by the lateral loads. In addition, there is only load-displacement curve in the results section, please supplement and description in combination with the corresponding testing phenomenon.
  • Line 368 ”To conclude the tension testing, a tensile force was applied until rupture, which occurred at 300 kN” However, the load displacement curve in figure 7 shows that the maximum value of Y-axis is 250 kN.
  • In 3.5.2 section, line 389 ” The following stiffness analysis includes an error in every stiffness data obtained due to the machine deformation throughout the test.” What kind of deformation of the test device caused the error? What is the error value? Please be more specific.

Reviewer 2 Report

This paper presents an automated connection solution to maximise the benefits of off-site manufacturing in multi-story buildings. The topic of the paper is interesting and can be published after addressing some concerns. However, the major concern regarding this paper is the flow of the paper. Many sections have been presented in the wrong places; for example, I can see the problem statement has been discussed in the result! So the authors should significantly revise the structure of the paper.
•    In the abstract, “However, off-site manufacturers focus their operations mainly in single-home construction.” It is a bit unclear for a reader that read it for the first time. Please clarify it.
•    The literature review in this paper is a bit weak. Although many papers in the literature can directly or indirectly be related to this study, many of them have been missed. I suggest that the authors divide section 1 into two sections, including the introduction and the literature review.
•    In the introduction, the research gaps and the contributions of the study should be highlighted.
•    It is good to see that the authors provided a comprehensive flowchart for the research methodology, but it is a bit messy. Please enhance the presentation of this figure. It is a bit hard for a reader to follow the different steps of this figure.
•    Section 2 is not well organised. Please check if it is possible to break the content of this section into several subsections.
•    Some staff related to the research problem have been discussed in section 3. While the research problem first should be clearly discussed before the research method, and then a research method should be developed to tackle the problem.

Author Response

Please see the attachment, the second part is addressed to you (Reviewer 2). 

Round 2

Reviewer 1 Report

the reviewer have no further comments.

Reviewer 2 Report

Accept